

# Determining the sensitive parameters of WRF model for the prediction of tropical cyclones in the Bay of Bengal using Global Sensitivity Analysis and Machine Learning

Harish Baki[1], Sandeep Chinta[1], C Balaji[1,2], and Balaji Srinivasan[1]

[1]Department of Mechanical Engineering, Indian Institute of Technology Madras, Chennai 600036, India
[2]Center of Excellence in Atmospheric and Climate Sciences, Indian Institute of Technology Madras, Chennai 600036, India

**Correspondence:** C Balaji (balaji@iitm.ac.in)

**Abstract.** The present study focuses on identifying the parameters from the Weather Research and Forecasting (WRF) model that strongly influence the prediction of tropical cyclones over the Bay of Bengal (BoB) region. Three global sensitivity analysis (SA) methods, namely the Morris One-at-A-Time (MOAT), Multivariate Adaptive Regression Splines (MARS), and surrogate-based Sobol' are employed to identify the most sensitive parameters out of 24 tunable parameters corresponding to seven

parameterization schemes of the WRF model. Ten tropical cyclones across different categories, such as cyclonic storms, severe cyclonic storms, and very severe cyclonic storms over BoB between 2011 and 2018, are selected in this study. The sensitivity scores of 24 parameters are evaluated for eight meteorological variables. The parameter sensitivity results are consistent across three SA methods for all the variables, and 8 out of the 24 parameters contribute $80\% - 90\%$ to the overall sensitivity scores. It is found that the Sobol' method with Gaussian progress regression as a surrogate model can produce reliable sensitivity results

when the available samples exceed 200. The parameters with which the model simulations have the least RMSE values when compared with the observations are considered as the optimal parameters. Comparing observations and model simulations with the default and optimal parameters shows that predictions with the optimal set of parameters yield a 16.74% improvement in the 10m wind speed, 3.13% in surface air temperature, 0.73% in surface air pressure, and 9.18% in precipitation predictions compared to the default set of parameters.

# 1  Introduction

The Indian subcontinent is vulnerable to tropical cyclones which develop in the North Indian Ocean (NIO) that consists of the Arabian Sea and the Bay of Bengal (BoB). These cyclones invariably cause widespread destruction to life and property. During the pre-monsoon and post-monsoon seasons, the tropical cyclones develop and bring heavy rainfall and gusts of wind towards the coastal lands (Singh et al., 2000). The number of tropical cyclones that form in the NIO has increased significantly during

the past few years, specifically during the satellite era (1981–2014). The frequency and duration of very severe cyclones in the BoB were increasing at an alarming rate, which alone contributed to an overall increase in the NIO (Balaji et al., 2018). An extensive study conducted using the past 30 years of data suggests that the severity of extremely severe cyclonic storms (ESCS) over NIO increased by 26%. The observed statistics reveal that the duration of the ESCS stage and maximum wind





speeds of ESCSs have shown an increasing trend, and the land falling category was very severe. Singh et al. (2021a). On
considering climate change, Singh et al. (2019) showed that present warming climate impacts on the formation and severity
of tropical cyclones over the BoB region. This ultimately affects the densely populated coastal cities adjacent to BoB, such
as Chennai, Visakhapatnam, Bhubaneswar, and Kolkatta (Singh et al., 2019). Reddy et al. (2021) showed that projecting the
present global warming conditions and climate changes into the near future leads to the intensification of the tropical cyclones
with ESCS and VSCS categories. Consequently, accurate predictions of cyclone track, landfall, wind, and precipitation are
critical in minimizing the damage caused by the tropical cyclones that are increasing in number and intensity.

The Weather Research and Forecast (WRF) model (Skamarock et al., 2008) is a community-based Numerical Weather
Prediction (NWP) system, which has been widely used to predict cyclones to date. The accuracy of the WRF model depends
on (i) the specification of initial and lateral boundary conditions, (i) the representation of model physics schemes, and (iii)
the specification of parameters. With the availability of vast computational resources and observations, the accuracy in the
specification of initial and lateral boundary conditions is improved to a great extent (Mohanty et al., 2010; Singh et al., 2021b).
Many researchers have studied the sensitivity of physics schemes in simulating tropical cyclones over the BoB and invariably
reported the performance of different combinations of physics schemes by comparing the tracks and intensities of cyclones
(Pattanayak et al., 2012; Osuri et al., 2012; Rambabu et al., 2013; Kanase and Salvekar, 2015; Chandrasekar and Balaji, 2016;
Sandeep et al., 2018; Venkata Rao et al., 2020; Mahala et al., 2021; Singh et al., 2021b; Messmer et al., 2021). However,
systematic studies on parameter sensitivity, to determine their optimal values is yet to be explored for tropical cyclones over
the BoB region.

Model parameters are the constants or exponents written in physics equations set up by the scheme developers, either through
observations or theoretical calculations. In some cases, the default parameters are selected based on trial-and-error methods.
This implies the parameters values may vary depending on the climatological conditions (Hong et al., 2004; Knutti et al.,
2002). The WRF model consists of a bundle of physics schemes, and there exist as many as a hundred tunable parameters
(Quan et al., 2016). Calibration of all the parameters to reduce the model prediction error is highly challenging, and it brings
several obstacles. First, a vast number of model simulations are required to perform parameter optimization, and the order goes
beyond $10^4$ with an increase in parameter dimension. Second, the WRF model can simulate various meteorological variables,
and each parameter may influence more than one variable. Thus, the parameter optimization needs to consider several variables
at once, which increases the computation cost even further (Chinta and Balaji, 2020). With the current situation and availability
of computational resources keeping in mind, performing thousands of numerical simulations for long periods such as tropical
cyclones is extremely expensive. The best remedy is to use sensitivity analysis to identify the parameters that significantly
impact the model simulation thereby reducing order of parameter dimension.

Sensitivity analysis is the method of uncertainty estimation in model outputs contributed by the variations in model inputs
(Saltelli, 2002). Several researchers (Green and Zhang, 2014; Quan et al., 2016; Di et al., 2017; Ji et al., 2018; Wang et al.,
2020; Chinta et al., 2021) have conducted sensitivity analysis of a number of parameters using various methods in the WRF
model. Green and Zhang (2014) studied the sensitivity of four parameters in the WRF model to the intensity and structure of
Hurricane Katrina, using single and multi-parameter designs, and reported that two parameters significantly affect the intensity





and the structure. Quan et al. (2016) examined the sensitivity of 23 adjustable parameters of the WRF model to 11 atmospheric

variables for the simulations of 9 five-day summer monsoon heavy precipitation events over the Greater Beijing, using the Morris One-at-A-Time (MOAT) method. The results showed that 6 out of 23 parameters were sensitive to most variables and five parameters were sensitive to specific variables. Di et al. (2017) conducted sensitivity experiments of 18 parameters of the WRF model to the precipitation and surface temperature, for the simulations of 9 two-day rainy events and nine two-day sunny events, over Greater Beijing. The authors have adopted four sensitivity analysis methods, namely the delta test, the sum

of trees, Multivariate Adaptive Regression Splines (MARS), and the Sobol' method. The results showed that five parameters greatly affected the precipitation, and two parameters affected surface temperature. Ji et al. (2018) investigated the sensitivity of 11 parameters to the precipitation and its related variables using the WRF model, for the simulations of a 30-day forecast, over China. The MOAT and surrogate-based Sobol' methods for the sensitivity analysis were used, and it was seen that the Gaussian Process Regression (GPR) based Sobol' method was found to be more efficient than the MOAT method. The results

also showed that four parameters significantly affect the precipitation and its associated quantities. Wang et al. (2020) studied the sensitivity of 20 parameters to various meteorological and model variables, for 30-day simulations, over the Amazon region. The MOAT, MARS, and surrogate-based Sobol' methods for sensitivity analysis were employed, the results showed that the three methods were consistent, and six out of twenty parameters contribute to $80\% - 90\%$ of the total variance. Chinta et al. (2021) studied the sensitivity of 23 parameters to eleven meteorological variables for the simulations of twelve 4-day

precipitation events during the Indian Summer monsoon, using the WRF model. The sensitivity analysis was conducted using the MOAT method with ten repetitions, and the results showed that 9 out of 23 parameters have a considerable impact on the model outputs. These studies show that hundreds of numerical simulations are required to perform sensitivity analysis. Thus, while selecting the sensitivity analysis methods and the number of parameters, the computational coast is a critical factor to consider.

Razavi and Gupta (2015) extensively studied the impact of numerous sensitivity analysis methods and reported that each method works based on a different set of ground-level definitions. The results from these methods do not always coincide. The studies proposed that while selecting a global sensitivity analysis method, one needs to consider four important characteristics, namely (i) local sensitivities, (ii) the global distribution of local sensitivities, (iii) the global distribution of model responses, and (iv) the structural organization of the response surface. The studies also reported that relying on only one sensitivity

analysis method may not yield feasible results since one single method may not be able to bring out all the characteristics fully. From these studies, one can infer that more than one SA method needs to be explored to improve confidence in the results obtained from sensitivity studies. The objective of the present study is to assess the sensitivity of the WRF model parameters to various meteorological variables such as surface pressure, temperature, wind speed, precipitation, and model variables such as radiation fluxes and boundary layer height, for the simulations of tropical cyclones over the BoB region, using three different

global sensitivity analysis methods.

This paper is organized as follows: a brief description of sensitivity analysis methods is presented in section 2. Section 3 presents the design of numerical experiments and sensitivity experimental setup. Section 4 shows the results of the three





sensitivity analysis methods and a comparison between simulations and observations, and section 5 gives the summary and conclusions.

## 2  Sensitivity Analysis Methods

Sensitivity analysis is the assessment of uncertainties in model outputs that are attributed to the variations in inputs factors (Saltelli et al., 2008). The sensitivity analysis proceeds as follows: (1) selecting the right model and corresponding best set of physics schemes, (2) identifying the adjustable input parameters and corresponding ranges, (3) choosing the sensitivity analysis methods, (4) running the design of experiments to generate the sample set of input parameters and running the model using these parameter sets, and (5) analyzing the model outputs obtained by different parameter samples and quantifying the sensitivity of selected parameters.

Sensitivity analysis methods are classified as derivative-based, response-surface-based, and variance-based approaches Wang et al. (2020). In mathematical terms, the change of an output concerning the change in the input is referred to as the sensitivity of that input, which is the principle of derivative-based SA. The Morris One-at-A-Time (MOAT) is a derivative based SA method(see subsection 2.1). The response-surface-based approach works on the differences between the responses of a mathematical model with all the input factors against that built with all but a particular input factor. The Multivariate Adaptive Regression Splines (MARS) method comes under this category (see subsection 2.2). For the variance-based approaches, the sensitivity of an input variable is defined as the contribution of the variance caused by the variable in question to the total variance of the model output. In mathematical terms, if the model output variance is decomposed by the contributions of each individual and combined interactions, then the highly sensitive factors will have a more significant variance contribution. The Sobol' sensitivity analysis comes under the variance-based approach (see subsection 2.3). The MOAT method requires a uniform space-filling design, whereas the MARS and Sobol' methods require random space-filling designs. The MOAT and MARS methods give a more qualitative analysis, whereas the Sobol' method gives a quantitative analysis (Wang et al., 2020). As already stated by Razavi and Gupta (2015), unique sensitivity analysis methods for all applications are scarce in the literature. Furthermore, they observed that using more versatile SA methods could improve the confidence in sensitivity results by compensating for the drawbacks of the individual SA methods. Thus, in the present study, three widely used SA methods are selected for sensitivity analysis because of the differences in their methodology, as a consequence of which the parameters that are sensitive to the numerical model are studied. One can then extract those parameters which turn out to be significant in all the methods under consideration, thereby bolstering the argument. These are the most influential parameters that need to be worked out to improve the forecast skill.

### 2.1  The MOAT method

The MOAT is a derivative-based sensitivity analysis method, also known as elementary effects method, which evaluates the parameter sensitivity according to the elemental effects of individual parameters (Morris, 1991). Consider a model with $n$ input parameters $X = (x_1, x_2, ... x_n)$ with variability in their ranges. The parameters are normalized to bound between





[0,1]. The parameter space is divided into $p$ equally dispersed intervals, which can be filled with the discrete numbers of $[0, \frac{1}{p-1}, \frac{2}{p-1}, ..., \frac{p-2}{p-1}, 1]$. Here $p$ is a user defined integer. An initial vector of input parameter $X^1 = (x_1^1, x_2^1, ... x_n^1)$ is randomly created by taking values from the defined parameter space. Following the One-at-A-Time method, one parameter is selected and perturbed by $\Delta$, i,e., $X_m^1 = (x_1^1, x_2^1, ..., x_m^1 \pm \Delta, ..., x_n^1)$. Here $\Delta$ is a randomly selected multiple of $\frac{1}{p-1}$. The model is run using these initial and perturbed vectors, and the elemental effect of of that parameter is calculated as:

$$EE_m^1 = \frac{f(X_m^1) - f(X^1)}{\Delta} \tag{1}$$

The subscript $m$ implies the $m^{th}$ parameter is perturbed and the superscript 1 is the indication of $1^{st}$ MOAT trajectory. In a single trajectory, this process is repeated for all parameters to compute the elementary effects of every parameter. The entire trajectory is replicated $r$ times randomly to obtain the reliable sensitivity results. At the end of the process, a total of $r \times (n+1)$ model simulations are evaluated to complete the MOAT sensitivity analysis. A modified mean of $|EE_m|$, $\mu_m$, and the standard deviation of $|EE_m|$, $\sigma_m$ are constructed as the sensitivity indices of input parameter $x_m$, as given below

$$\mu_m = \sum_{i=1}^{r} \frac{|EE_m^i|}{r} \tag{2}$$

$$\sigma_m = \sqrt{\frac{\sum_{i=1}^{r} (EE_m^i - \mu_m)^2}{r}} \tag{3}$$

A high value of $\mu_m$ implies that the parameter $x_m$ has a more significant impact on the model output. In contrast, a high value of $\sigma_m$ indicates the nonlinearity of $x_m$ or high interactions with other parameters.

## 2.2 The MARS method

MARS is an extension of Recursive Partition Regression model with the ability of continuous derivative (Friedman, 1991). The model is constructed by a forward and a backward passes: the forward pass divides the entire domain into a number of partitions and a overfitted model is produced by localized regressions in every partition, and the backward pass prunes the overfitted model to a best model by repeatedly removing least concerned basis function at a time. The MARS model can be decomposed as:

$$\hat{f}(x) = a_0 B_0 + \sum_{m=1}^{M} \sum_{\substack{K_m=1 \\ i \in V(m)}} a_m B_m(x_i) + \sum_{m=1}^{M} \sum_{\substack{K_m=2 \\ (i,j) \in V(m)}} a_m B_m(x_i, x_j) + ... \tag{4}$$

The basis functions can be a constant $(B_0)$, a hing function $(B_m(x_i))$, or a product of two or more hing functions $(B_m(x_i, x_j))$. The coefficients $(a_0, a_1, ..a_m)$ are determined by linear regression in every partition. The Generalized Cross Validation (GCV) score of every model during the backward pass is calculated as:

$$GCV(m) = \frac{1}{N} \frac{\sum_{i=1}^{N} [y_i - \hat{f}_m(X_i)]^2}{[1 - \frac{C(m)}{N}]^2} \tag{5}$$





Here $N$ is the number of samples before pruning, $y_i$ is the target data point, $\hat{f}_m(X_i)$ is the $m^{th}$ model estimated data point corresponding to the input data $X_i$, and $C(m)$ is the penalty factor accounted for the increase in variance due to the increase in complexity. The difference between the GCV scores of the pruned model with the over-fitted model is measured as the

importance of that parameter that has been removed. This implies that a higher difference indicates a higher sensitivity of that parameter.

### 2.3 The Sobol' method

The Sobol' sensitivity analysis works on the basis of variance decomposition (Sobol, 2001). Consider a response function $f(x)$ of a random vector $x$. The ANalysis Of VAriance (ANOVA) decomposition of $f(x)$ is written as:

$$f(x) = f_0 + \sum_{1<i<n} f_i(x_i) + \sum_{1<i,j<n} f_{ij}(x_i,x_j) + ... + f_{12...n}(x_i,x_j,...,x_n) \tag{6}$$

The variance of $f(x)$ can be expressed as the contributions of variance of each term in the equation (6), i,e.,

$$\int f^2(x)dx - f_0 = \sum_{1<i<n} \int f_i^2(x_i)dx_i + \sum_{1<i,j<n} \int f_{ij}^2(x_i,x_j)dx_idx_j + ...$$
$$+ \int f_{12...n}^2(x_i,x_j,...,x_n)dx_1dx_2...dx_n \tag{7}$$


$$D = \sum_{1<i<n} D_i + \sum_{1<i,j<n} D_{ij} + ... + D_{12...n} \tag{8}$$

Where $n$ is the total number of parameters, $D$ is the total variance of output response function, $D_i$ is the variance of $x_i$, $D_{ij}$ is the variance of interactions of $x_i$ and $x_j$, and $D_{12...n}$ is the variance of interactions of all parameters. The Sobol' sensitivity indices of a particular parameter are defined as the ratio of individual variances to the total variance, and these can be written

as:

$$S_i = \frac{D_i}{D}; \quad S_{ij} = \frac{D_{ij}}{D}; ... \quad \text{and} \quad S_{12...n} = \frac{D_{12...n}}{D} \tag{9}$$

These indices explain the effects of first order, second order, and total order interactions, respectively. From equations (8) and (9) it is evident that the sum of all the indices is equal to 1. Finally, the total order sensitivity index of $i^{th}$ parameter can be calculated as the sum of all the interactions of that parameter, i,e.,

$$S_{T_i} = S_i + S_{\substack{ij \\ i \neq j}} + ... + S_{123...i...n} \tag{10}$$

Generally, while the computation of first and second order effects is rather straight forward, the calculation of higher order effects is very expensive because the dimension of the higher order terms is very large. To solve this problem, Homma and Saltelli (1996) introduced a new total sensitivity index as

$$S_{T_i} = 1 - \frac{D_{-i}}{D} \tag{11}$$





Where $D_{-i}$ indicates the total variance of response function without the consideration of the effects of the $i^{th}$ parameter. A higher total order sensitivity index implies higher importance of that parameter.

## 3  Design of numerical experiments

### 3.1  WRF model configuration and adjustable parameters

In the present study, the Advanced Research WRF (WRF-ARW) model version 3.9 (Skamarock et al., 2008) is used for the
numerical experiments. The model consists of two domains, d01 and d02, correctly aligning at the center, with a horizontal resolution of 36 km and 12 km. The inner domain, which is our area of interest, consists of $360 \times 360$ grid points that encapsulate the BoB and cover the Indian subcontinent along with the northern Indian Ocean. The outer domain consists of $240 \times 240$ grid points and is kept reasonably away from the inner domain. The simulation domains are illustrated in Figure 1. The model consists of 50 terrain-following $\sigma$ layers in the vertical direction, while the top layer is kept at 50 hPa. The model is
integrated with a time step of 90 sec and 30 seconds for domains d01 and d02, respectively. The NCEP FNL (National Centers for Environmental Predictions) operational global analysis and forecast data at $1° \times 1°$ resolution with a six-hourly interval (National Centers for Environmental Prediction/National Weather Service/NOAA/U.S. Department of Commerce. updated daily., 2000) are provided as the initial and lateral boundary conditions for the simulations. The simulations are carried out for 108 hours, including 12 hours of spin-off time.

Parameterization schemes represent the physical processes that are unresolved by the WRF model. The WRF model consists of seven different parameterization schemes: microphysics, cumulus physics, short wave and longwave radiation, planetary boundary layer physics, land surface physics, and surface layer physics. The parameterization schemes used in this study are: rapid radiative transfer model (Mlawer et al., 1997) for longwave radiation, Dudhia shortwave scheme (Dudhia, 1989) for shortwave radiation, revised MM5 scheme (Jiménez et al., 2012) for surface layer physics, Unified Noah land surface model
(Mukul Tewari et al., 2004) for land surface physics, Yonsei University Scheme (YSU) (Hong et al., 2006) for planetary boundary layer physics, Kain-Fritsch (Kain, 2004) for cumulus physics, and WRF Single-Moment 6-class (WSM6) scheme (Hong and Lim, 2006) for microphysics. A total of 24 tunable parameters are selected based on the guidance from literature (Di et al., 2015; Quan et al., 2016). The list of parameters and corresponding ranges are presented in Table 1. Though the selected parameter may not cover the entire existing parameters, the availability of computational resources limits the experimental
design. The experimental design is based on the most critical parameters that are more likely to significantly influence the model output.

### 3.2  Simulation events, WRF model output variables, and observational data

In the present study, ten tropical cyclones that originated in the Bay of Bengal during the period of 2011 to 2017 are selected for the numerical experiments. The cyclones are chosen from various categories to generalize the experiments to ensure the
robustness of the outcomes. The Indian Meteorological Department (IMD) categorizes the cyclones based on the Maximum





Sustained surface Wind speed (MSW) for a three-minute duration. The tropical cyclone categories used in this study are Cyclonic Storm (34-47 Knots), Severe Cyclonic Storm (48-63 Knots), and Very Severe Cyclonic Storm (64-119 Knots) (Srikanth et al., 2012). Figure 2 illustrates the IMD observed tracks of selected cyclones, with a clear indication of their category. Table 2 presents the details of category, landfall time, and the simulation duration of the cyclones selected in the present study. Each
cyclone is simulated for 108 hours, including 12 hours of spin-off time, 72 hours of simulation before the landfall, and 24 hours of simulation after the landfall. The sensitivity of parameters is conducted for different meteorological variables: wind speed 10 meters above ground(WS10), temperature 2 meters above ground (SAT), surface pressure (SAP), total precipitation (RAIN), planetary boundary layer height (PBLH), outgoing longwave radiation flux (OLR), downward short wave radiation flux (DSWRF), and downward longwave radiation flux (DLWRF). The WRF simulations of these variables are stored at 6-hour
intervals.

    The simulations are validated against the Indian Monsoon Data Assimilation and Analysis (IMDAA) data (Ashrit et al., 2020) and Integrated Multi-satellitE Retrievals for GPM (IMERG) dataset (Huffman, G and Savtchenko, AK, 2019). The IMDAA data is available at $0.12° \times 0.12°$ resolution with a six-hour latency and the IMERG data is available at $0.1° \times 0.1°$ resolution with a thirty-minute latency. Since the model resolution is close to the validation data resolution, it results in very
little or no loss of data after regridding takes place. The accumulated precipitation data for validation is taken from IMERG data, while the remaining variables are taken from IMDAA data. Apart from this data, the maximum sustained wind speed (MSW) observations at the storm center for every cyclone, provided by the IMD at 3-hour intervals, are also used for validation.

### 3.3   Experimental setup

The sensitivity analysis requires a large set of values of the parameters assigned to the WRF model, following which simulations
are performed. Uncertainty Quantification Python Laboratory (UQ-PyL) is an uncertainty quantification platform, designed by Wang et al. (2016), which is used to generate the parameter samples for the MOAT method. Based on the studies of Quan et al. (2016), the parameter samples are generated with $p = 4$ and $r = 10$, which yields a total of $10 \times (24 + 1) = 250$ parameter samples, for the selected 24 parameters. These parameter sets are assigned in the WRF model, and a total of $250 \times 10 = 2500$ simulations are performed across ten cyclones. Once the simulations are completed, the output meteorological variables are
extracted and stored at six-hour intervals. The sensitivity indices for all the parameters are calculated based on equations (2) and (3) which are implemented in the UQ-PyL, and the indices averaged over all the cyclones to generalize the results.

    In contrast, the MARS and Sobol' methods require a different set of samples compared to the MOAT method. Based on the previous studies (Ji et al., 2018; Wang et al., 2020), the quasi-Monte Carlo (QMC) Sobol' sequence design (Sobol', 1967) is employed to create 250 parameter samples, using UQ-PyL package for each event. Similar to the MOAT method, these param-
eter samples are assigned in the WRF model, and another 2500 simulations are performed for the cyclones under consideration. The output variables are extracted and stored at 6-hour intervals. The evaluation of sensitivities using the MOAT method requires simulations only from the WRF model. In contrast, the MARS and Sobol' methods require skill score metrics between the simulation and observations. In the present study, the RMSE score between simulation and observation is employed as the





skill score metric, which is formulated as

$$RMSE = \sqrt{\frac{\left[\sum_{l=1}^{L}\sum_{k=1}^{K}\sum_{j=1}^{J}\sum_{i=1}^{I}(sim_{ijkl} - obs_{ijkl})^2\right]}{I \times J \times K \times L}} \qquad (12)$$

Where $I$ and $J$ are the number of grid points in lateral and longitudinal direction, $K$ is the dimension of times, $L$ is the number of cyclones, *sim* is the simulated value, and *obs* is the observed value. Since the same parameter set is employed for all the cyclones, equation (12) is employed to get one RMSE value corresponding to one parameter sample. The parameter set and RMSE are given as inputs and targets to the MARS solver, and the MARS sensitivity indices are computed following GCV equation (5).

The Sobol' method, as a quantitative sensitivity analysis method, gives more accurate and robust results, albeit at a much higher computational cost. The Sobol' method may require $[10^3 \sim 10^4 \times (n+1)]$ (i.e., $n$ is the number of parameters) number of model runs to get accurate results. This is exceedingly challenging even if supercomputing facilities are available. To circumvent this difficulty, one can use the surrogate models instead of running the WRF model for more simulations. The surrogate models are powerful machine learning tools that can correlate the empirical relations between inputs (i.e., parameter set) and the targets (i.e., RMSE matrix). In the present study, five different surrogate models namely Gaussian Process Regression (GPR)(Schulz et al., 2018), Support Vector Machines (SVM)(Radhika and Shashi, 2009), Random Forest (RF)(Segal, 2005), Regression Tree (RT)(Razi and Athappilly, 2005), and K Nearest Neighborhood (KNN)(Rajagopalan and Lall, 1999) are selected for evaluation. The surrogate models are provided with the parameter set as inputs and the RMSE as the target, and the models are trained on this data. The goodness of fit is considered as the accuracy metric, which is calculated as,

$$R^2 = 1 - \frac{\sum_{i=1}^{N}(\hat{y}_i - y_i)^2}{\sum_{i=1}^{N}(y_i - \overline{y}_i)^2} \qquad (13)$$

Where $N$ is the total number of samples, $y_i$ is the true value, $\hat{y}_i$ is the predicted value, and $\overline{y}$ is the mean of true values. The accuracy of the surrogate models is examined by applying ten-fold cross-validation, which is implemented as follows. The entire data is divided into ten equally spaced subsets. The data in $k^{th}$ fold is kept as the test set, whereas the data from the remaining folds is taken as the training set. The surrogate model gets trained on this training set, and the predictions corresponding to the test set are estimated. This procedure is iterated for all folds, and the predictions of all folds are stacked into one set. This way, an entire prediction set corresponding to the test set is generated. These two sets are provided as predictions, and true values to equation (13), and the goodness of fit ($R^2$) is calculated. The surrogate model with the highest $R^2$ value is selected as the best model. Once the best surrogate model is attained, the Sobol' sequence is used to generate 50000 parameter samples, and the surrogate model predicts the corresponding outputs. Based on these outputs, the sensitivity indices are calculated. Pedregosa et al. (2011) have implemented the MARS method, Sobol' method, and the selected five surrogate models in Python language under the scikit-learn module, as Application Programming Interfaces (API). The APIs of sensitivity methods and surrogate models are used in the present study.





## 4   Results and Discussion

### 275   4.1   MOAT sensitivity analysis

The sensitivity indices of parameters corresponding to the selected meteorological variables are calculated based on the MOAT method. The modified means of each variable under consideration are normalized to the range of [0,1]. They are illustrated as a heatmap in Figure 3, with a darker shade indicating the highest sensitivity and a lighter shade indicating the least sensitivity. Figure 3 shows that parameter P14 has the highest sensitivity to most of the variables, followed by parameter P6. The parameters
P3, P4, P10, P15, P17, P21, and P22 also show high sensitivity to at least one of the variables. In contrast, the parameters P1, P8, P11, P13, P16, P18, and P20 seem insensitive to any one of the variables, and the remaining parameters have a minimal contribution. A close observation of Figure 3 reveals that the variables OLR and DSWRF having the highest sensitivity to just one parameter each, whereas the remaining variables exhibit the highest sensitivity to at least two parameters.

The uncertainties that lie in the sensitive parameters is examined by observing the distribution of the parameters. Since the
available data points are limited to only ten samples, a resampling method can be employed to procure more samples without further numerical model runs. The bootstrap resampling (Efron and RJ, 1993) is an efficient way to generate the same number of samples as the original dataset, with replacement allowed. In the present study, the bootstrap method is employed for 100 applications to generate ten samples with replacement. In this way, a new dataset of $(100 \times 10)$ is created for one parameter corresponding to one variable. The distribution of each parameter is illustrated as a boxplot in Figure 4. In this figure, for every
parameter, the horizontal red line inside the box indicates the median value, the upper and the lower bounds of the box are (mean $\pm$ one standard deviation value), and the upper and lower whiskers are the maximum and minimum values. The boxplot shows that the most sensitive parameters exhibit either a higher variance or a higher median value (Wang et al., 2020). For the variable OLR, Figure 4(f) shows that the parameter P10 has the highest median value with large variance, whereas the parameters P6 and P12 have the least median value with large variances. Figure 4(g) shows that parameter P14 has the highest
sensitivity to the variable DSWRF and has a very minimal variance, whereas the sensitivity of the remaining parameters is comparably very minimal. Figures 4(c,e,h) show that the variables SAP, PBLH, and DLWRF have more than three sensitive parameters. The results show that except for DSWRF and OLR, all the variables have at least two high sensitive parameters. The results obtained by the boxplot strengthen that of the heatmap results.

### 4.2   MARS sensitivity analysis

The GCV scores of 24 parameters corresponding to the selected variables are calculated based on the MARS method. Figure 5 illustrates the heatmap of normalized GCV scores, with 1 indicating the highest sensitivity and 0 indicating the least sensitivity. The intensity signatures of Figure 5 are very consistent with that of Figure 3. The results show that most of the variables are sensitive to P14, followed by Parameter P6. In addition to this, the parameters P3, P4, P10, P15, P17, and P22 are seen to affect at least one of the dependent variables. The results also reveal that P1, P2, P8, P11, P13, P16, P18, P19, P20, P21, and
P24 do not significantly influence any of the variables. A close observation of Figure 5 reveals that the variables WS10, OLR, and DSWRF are sensitive to only one parameter each. In contrast, variables SAP, PBLH, and DLWRF are influenced by more





than three parameters. The distribution of results is obtained by applying the bootstrap method, which is employed for 100 applications to generate 250 samples with replacement. In this way, a new dataset of $(100 \times 250)$ is created for one parameter corresponding to one variable. Figure 6 shows the boxplot of the MARS GCV scores generated by the bootstrap resampling dataset. Figures 6(a,f,g) show that the variables WS10, OLR, and DSWRF are sensitive only to one parameter each. Similarly, Figures 6(c,h) show that the variables SAP and DLWRF are sensitive to more than three parameters. The remaining variables are sensitive to at least two parameters. The results of the boxplot corroborate the results from the heatmap. These results are very consistent with that of the MOAT method.

### 4.3 Sobol' sensitivity analysis

The Sobol' method calculates the contribution of variation of the individual parameters to the total variance of the output by performing computations on a vast number of data. Thus, the Sobol' method is considered as a quantitative analysis, which produces more reliable results. Due to the limitation of computational resources, a large number of model simulations are impractical to perform. The best remedy is to use surrogate models as an alternative to the original model, which can be trained on the limited samples produced by the original model, as already briefly mentioned. This implies that the Sobol' method's accuracy relies critically on the accuracy of the surrogate model. Thus, it becomes imperative to validate the surrogate model before analyzing the sensitivity of the parameters.

#### 4.3.1 Validation of surrogate models

Figure 7 shows the distribution of $R^2$ scores of different surrogate models for the selected meteorological variables by applying bootstrap resampling. In Figure 7, each subplot corresponds to one meteorological variable, and the horizontal and vertical axes indicate the surrogate models and the goodness of fit $(R^2)$ value, respectively. For every meteorological variable, the GPR model has the highest $R^2$ value, which is close to 1, and the variance is also minimal. This implies that the GPR model can accurately correlate the empirical relations between inputs and outputs. In contrast, the remaining surrogate models show high variance in respect of at least one of the variables. Figure 7(c) shows that the regression tree has the highest variance with the least $R^2$ value, and the minimum whisker lies below zero, which indicates the inability of the RT in capturing the correlations. In every subfigure, the $R^2$ value of KNN is close to 0.5, which implies that the model can explain only 50% of the total variance around its mean. The surrogate models SVM and RF have very close accuracy except for the variable ORL, in which the SVM shows high variance with $R^2$ value close to 0.5. These results indicate that all the remaining models have inconsistencies in their accuracy except for the GPR model. Figure 8 shows a scatter plot of the WRF model output against the GPR fit output for the eight variables under consideration. In Figure 8, each subplot corresponds to one meteorological variable, and the horizontal and vertical axes indicate the output of the WRF model and GPR fit, respectively. From the $R^2$ value shown in the plots, it is clear that the GPR model can explain 95% of the variability of the output data around its mean, except for the variable surface pressure, for which the $R^2$ value is 0.88 (as shown in Figure 8(c)). In view of the above, the GPR is chosen as the best surrogate model for the sensitivity studies with Sobol'.





### 4.3.2 Effects of sample size on surrogate model accuracy

The accuracy of a surrogate model depends on the number of samples provided to the model. At the same time, the sample size determines the computational cost required to perform additional model simulations. Thus, one needs to identify the minimum number of samples on which the surrogate model can attain reasonable accuracy. The effects of sample size on GPR's accuracy are evaluated as follows. The original dataset is divided into five sets: 50, 100, 150, 200, and 250 samples. Each set is bootstrap resampled for 100 instances, on which the accuracy of the GPR model is evaluated using ten-fold cross-

validation. The distribution of $R^2$ values for different sample sizes are illustrated in Figure 9, in which each subplot corresponds to one meteorological variable. The abscissa and the ordinate indicate the number of samples and $R^2$ value, respectively. From this figure, it is evident that the accuracy of the GPR model increases monotonically with an increase in the sample size. The $R^2$ value has high variance at 50 and 100 sample sizes, whereas there is minimal variance found at 200 and 250 sample sizes. It is found that the sample sizes 200 and 250 have identical $R^2$ values, and there is little improvement found by increasing the

samples beyond 200. Hence, based on the above results, it can be concluded that 200 samples are sufficient to construct a GPR model with adequate accuracy. Since the available data has 250 samples, the GPR model constructed with 250 samples is used in the Sobol' analysis.

### 4.3.3 Results of surrogate-based Sobol' method

The GPR model, which is built upon 250 samples, is used to predict the outputs of 50000 samples generated by Sobol'
sequence, and these outputs are used to estimate the Sobol' sensitivity indices, corresponding to each variable. Figure 10 illustrates the heatmap of normalized total order sensitivity indices, with 1 indicating the highest sensitivity and 0 indicating the least sensitivity, which is very consistent with the results of MOAT and MARS method shown in Figures 3 and 5 earlier. The parameter P14 is seen to be the most influential, followed by parameter P6. At least one of the dependent variables is sensitive to parameters P3, P4, P10, P15, P17, and P22. Comparing the results of Sobol' method with MOAT and MARS methods, it is
seen that the sensitivity patterns of each variable are showing consistency.

Figures 11(a-h) show the detailed illustration of the sensitivity indices of each meteorological variable. In each subfigure, the blue bar shows the first-order (primary) effects, the red bar shows the higher-order (interaction) effects, and the sum of these two show the total order effects. The advantage of Sobol' method is that the method can provide quantification of interaction effects. Figures 11(c,e,f) show that the SAP, PBLH, and OLR have considerable higher-order effects, which indicate that the
interactions are predominate in these variables. Figure 11(b,g) show that the variables SAT and DSWRF have only one sensitive parameter each, while Figures 11(c,e,h) show that the variables SAP, PBLH, and DLWRF are influenced by more than three parameters. These results strengthen the analogy obtained through the heatmap.

The results from Sobol' method indicate that only a few parameters contribute much to the sensitivity of the output variables. Figure 12 shows the aggregate relative contribution of total order effects of each parameter, corresponding to the selected
variables. The abscissa indicates the output variables, and the ordinate indicates the relative importance, and the parameters are indicated by different colors. The figure shows that only 8 out of 24 parameters, namely: P3, P4, P6, P10, P14, P15, P17, and





P22, are responsible for more than 80% of the total sensitivity of every variable. Unlike MOAT or MARS methods, Sobol's deterministic nature gives more accurate results as they are free of any uncertainties.

### 4.4    Physical interpretation of parameter sensitivity

The results obtained by the three sensitivity analysis methods suggest that only a few parameters strongly influence the meteorological variables under consideration in this study. The sensitivity indices of parameters obtained by the three methods are added over all variables and are normalized to $[0, 1]$. The results are shown in Figure 13 in descending order, which indicates the ranks of the parameters. Figure 13 shows that eight parameters: P3, P4, P6, P10, P14, P15, P17, and P22 strongly influence the variables combined. Additionally, there is a near-exact matching of all the three sensitivity methods, with little variation in

their ranks.

The results show that the parameter P14 is the most sensitive parameter among all. This represents the scattering tuning parameter used in the shortwave radiation scheme proposed by Dudhia (1989). This parameter is used in the downward component of solar flux equation (Montornès et al., 2015). This parameter is the main constant associated with the scattering attenuation and directly affects the solar radiation reaching the ground in the form of DSWRF. When a cloud is present in the

atmosphere, it attenuates the downward solar radiation; simultaneously, it contributes to the downward longwave radiation by means of multiple scattering. Since the Dudhia (1989) scheme does not have a representation of the multi-scattering process, parameter P14 attenuates the downward radiation without any contribution to the heating rate (Montornès et al., 2015). This leads to changes in the DLWRF. The land surface model transforms the solar radiation into other kinds of energies, such as latent heat (LH) and sensible heat (SH) near the surface. This implies that the changes caused to the downward radiation will

also affect the LH and SH. The planetary boundary layer is governed by the LH and SH. Therefore, the changes in the DSWRF will ultimately affect PBLH Montornès et al. (2015). A higher value of P14 leads to a decrease in downward solar radiation and the surface level heating, which ultimately reduces the surface atmosphere temperature (SAT). Studies of Quan et al. (2016) show that the changes in SAT lead to variations in relative humidity. Due to the correlation between SAT, humidity, and SAP, variations in SAT and humidity lead to variations in the SAP.

The parameter P6 is the multiplier for entrainment mass flux rate in the Kain-Fritsch cumulus physics scheme. This parameter determines the amount of ambient air entraining into the updraft flux, which further dilutes the updraft parcel. A high value of P6 indicates a high amount of ambient air entrainment into the air parcel, and the atmosphere becomes more stable. This suppresses the formation of deep convection, which leads to a decrease in convective precipitation. Since the shallow convection removes a large amount of the instabilities, this leads to more stable stratiform clouds, ultimately resulting in high precipitation.

The occurrence of precipitation decreases the SAT and increases the relative humidity, leading to a change in the SAP. This parameter alters the formation of clouds, which in turn affects the variables that depend on clouds, such as OLR, DSWRF, and DLWRF (Quan et al., 2016; Ji et al., 2018).

The parameter P17 is the multiplier of saturated soil water content used in the Unified Noah Land Surface scheme, proposed by Mukul Tewari et al. (2004). The saturated soil water content plays a prominent role in heat exchange between land and

surface through moisture transportation in soil and evaporation. The SAT is affected by the amount of evaporation at the





surface, which implies that changes in parameter P17 lead to SAT variations. The sensible heat and the latent heat are the two prime modes of heat exchange at the surface and the evaporation, on which the PBLH depends. Thus, parameter P17 also affects PBLH. Evaporation is the main constituent of cloud formation. Since the parameter P17 affects evaporation, the DLWRF, which depends on clouds, will also be affected by P17.

The parameter P10 is the scaling factor applied for icefall velocity used in the microphysics scheme, proposed by Hong et al. (2006). This parameter controls the ice terminal fall velocity, which governs the sedimentation of ice crystals. The cloud constituents such as cloud water and cloud ice are affected by the sedimentation of ice crystals. Since the cloud water and cloud ice reflect radiation into the outer space, any change in the parameter P10 causes variations in the OLR (Quan et al., 2016; Di et al., 2017; Ji et al., 2018). The parameter P4 is the Von Kármán constant used in the surface layer scheme (Jiménez

et al., 2012) and PBL scheme (Hong et al., 2006). This parameter relates the flow speed profile in a wall-normal shear flow to the stress at the boundary. This parameter directly influences the bulk transfer coefficient of momentum, heat, moisture, and diffusivity coefficient of momentum. This implies the changes in P4 will bring implicit variations in surface pressure and moisture, which lead to changes in the precipitation Wang et al. (2020).

The parameter P22 is the profile shape exponent for calculating the moment diffusivity coefficient used in the PBL scheme.

This parameter is directly related to P4 since both are used in the diffusivity coefficient of the momentum equation. This parameter regulates the mixing intensity of turbulence in the boundary layer, and because of this, the planetary boundary layer height (PBLH) will be affected (Quan et al., 2016; Di et al., 2017; Wang et al., 2020). The parameter P15 is the diffusivity angle for cloud optical depth computation used in the longwave radiation scheme, proposed by Mlawer et al. (1997). The longwave radiation irradiating back to the Earth's surface is attenuated by the diffusivity factor (which is the inverse of cosine

of diffusivity angle) multiplied by the optical depth. Thus, changes in P15 directly cause variations in DLWRF (Quan et al., 2016; Di et al., 2017; Iacono et al., 2000; Viúdez-Mora et al., 2015). The parameter P3 is the scaling factor for surface roughness used in the surface layer scheme (Jiménez et al., 2012). A smooth surface lets the flow be laminar, whereas a rough surface drags the flow, thereby affecting the near-surface wind speed (Nelli et al., 2020). This way, parameter P3 is directly related to the wind speed. Thus, any changes in P3 results will also affect the surface wind speed (Wang et al., 2020).

## 430  4.5    A comparison between simulations with the default and optimal parameters

The objective of the present work is to identify the most important parameters which greatly influence the model output variables. In the present study, the parameters with which the model simulations show the least RMSE error with respect to the observations are selected as optimal parameters. However, these parameters can be further optimized by a procedure followed by Chinta and Balaji (2020) to improve the model predictions of output variables which are greatly affected by the parameters.

To illustrate whether parameter optimization can improve model prediction, a comparison of WRF simulations with the default and optimal parameters for the meteorological variables, such as precipitation, surface temperature, surface pressure, and wind speed, was conducted. The RMSE values of WS10, SAT, SAP, and precipitation of the default and optimal simulations are evaluated and are shown in Table 3. The results show that optimal simulations have smaller RMSE values for surface wind (2.11 m/s) compared to default simulations (2.53 m/s). The percentage improvement is calculated as the percentage of





reduction in RMSE score between the default and optimal simulations over the default simulations. Table 3 shows that a 16.74% of improvement is achieved by using the optimal parameters over the default parameters for the simulations of surface wind speed. Similarly, the optimal parameters yield improvements of 3.13% for surface temperature, 0.73% for surface pressure, and 9.18% for precipitation, over the default parameters.

Taylor statistics (Taylor, 2001) is used to evaluate the accuracy of the model forecasts of WS10, SAT, SAP, and precipitation,
simulated with the default and optimal parameters. The Taylor statistics consists of centered Root-Mean-Square error, correlation coefficient, normalized standard deviation, and bias, which can be plotted in one Taylor as shown in Figure 14. The arcs centered at the origin represent the normalized standard deviation with the observed standard deviation located at the arc radius of 1. The simulation points close to the reference standard deviation arc imply that the variance in the simulations is similar to that of the observations. The arcs centered at the REF point on the abscissa represent the centered RMS error with the obser-
vations. The simulation points close to the REF point indicate that the RMS error between the simulations and observations is very minimal. The correlation coefficient is the cosine of the position vector of a point, with zero being least correlated and one being highest correlated with the observation. The bias is the difference between the means of simulations and observations, which is merely indicated by up or down triangles on the plot. The points close to the REF point indicate the highest correlation, variance close to observations, and least RMS error, implying best performance. The default and optimal simulations
of the SAT and SAP show no difference in any statistic implying the similar performance of the parameters. The optimal and default simulations of WS10 are positioned midway to the reference standard deviation arc on either side implying that the optimal simulations have less variance and the default simulations have more variance compared to the observations, whereas the standard deviation is same for both. Both the simulations lie on same correlation vector and on same semicircle originated from the REF point, implying that the simulations have same centered RMSE and correlation coefficients. The optimal simu-
lations of precipitation show less RMS error and high correlation compared to the default simulations. Even though the default simulations positioned close to the reference arc than the optimal simulations, the distance between the REF point and the optimal simulations is less compared to the default simulations, implying the best performance of the optimal parameters.

Figure 15 shows the domain averaged spatial distributions of the bias in the variables evaluated as the difference between the simulations with the default set of parameters and the observations on the left panel, and the difference between the simulations
with the optimal set of parameters and observations on the right. The IMDAA data is used to validate the WS10, SAT, and SAP, and the IMERG rainfall data is used to validate precipitation. For surface wind speed, Figure 15(a,b) show that the default simulations have large spatial coverage of 2 m/s, 3 m/s, and 4 m/s positive bias over the Bay of Bengal region, whereas the optimal simulations have 2 m/s bias over this region. The default and optimal simulations have similar spatial coverage over the land, whereas the optimal simulations show lesser bias compared to the default simulations. The surface plots clearly
show that the optimal parameters improved the surface wind speed simulations compared to the default parameters. For surface temperature, Figures 15(c,d) show that the default and optimal simulations have similar spatial distributions of temperature bias over the entire domain, with very minimal differences are observed over the northwest, southwest, and Bangladesh regions. Over these regions, the optimal simulations show a little less bias compared to the default simulations. For surface pressure, Figures 15(e,f) shows that the default and optimal simulations have similar spatial structures of bias over the entire domain





with seemingly no variations at all. Figure 15(g,h) show that the default simulations have larger spatial structures with higher bias compared to the optimal simulations over the north BoB, Bangladesh coast, south-east BoB, and central BoB regions. These results indicate that the optimization of the sensitive parameters with respect to wind speed and precipitation will yield more improvement.

The WRF model runs with optimal parameters improved the simulations of meteorological variables at the surface level. However, the optimal parameters indeed exert an impact on the upper atmospheric variables, and the performance of optimal parameters for the simulations of variables at this level should be satisfactory to use in the future. For this purpose, the wind fields at 500 hPa of vscs Thane and cyclone Phailin, simulated by the default and optimal parameters, are compared with observations, as shown in Figures 16 and 17. For cyclone Thane, at the end of day1, Figures 16(a1,b1,c1) show that the default and optimal parameters simulated similar cyclonic circulations and traces of anticyclonic circulations that are well matching with the observations. At the end of day2, Figures 16(a2,b2,c2) show that the optimal parameters simulated a well structured cyclonic circulation, whereas the default parameters simulated irregularities around the cyclonic circulation that were not observed. Both the parameters simulated an anticyclonic circulation with a spatial deviation to that of the observed one. At the end day3, Figures 16(a3,b3,c3) show that the default parameters simulated an anticyclonic circulation, but failed to simulate a cyclonic circulation. In contrast, the optimal parameters simulated a well structured cyclonic circulation with a spatial deviation and an anticyclonic circulation. For cyclone Phailin, at the end of day1, Figures 17(a1,a2,a3) show that the default and optimal parameters overestimated the cyclonic circulation intensity, however the optimal simulations show relatively less intensity than the default simulations. At the end of day2, Figures 17(a2,b2,c2) show that default and optimal simulations have similar intense cyclonic circulations at the observed location with an overestimation compared to the observations. At the end of the day3, Figures 17(a3,b3,c3) show that the optimal simulations have relatively similar intensity compared to the observations than the default simulations. These results show that the optimal parameters simulated the velocity field at 500 hPa with less intensity and close to the observations than the default parameters.

The Maximum Sustained Wind speed (MSW) is one of the primary measures of the intensity of a cyclone, and predicting an accurate MSW is of primordial importance for early warnings. In addition to the spatial distributions of variables, MSW is also compared for default and optimal simulations and shown in Figure 18. From the WRF simulations using QMC samples, MSW values of the ten cyclones are extracted at 6-hour intervals, beginning from the $18^{th}$ hour to the $84^{th}$ hour, and the data thus obtained is averaged over all the cyclones. A boxplot is generated using the data, and this shows that uncertainties in the parameters significantly affect the MSW simulations. The simulated MSW values with the default and optimal parameters are plotted along with the observed IMD MSW values, which show that the optimal simulations match quite well with the observations compared to the default simulations. These results indicate that the optimization of parameters will definitely improve model predictions.





# 5    Conclusions

The present study evaluated the sensitivity of the eight meteorological variables, namely surface wind speed, surface air temperature, surface air pressure, precipitation, planetary boundary layer height, downward shortwave radiation flux, downward longwave radiation flux, and outgoing longwave radiation flux, to 24 tunable parameters for the simulations of ten tropical

cyclones over the BoB region. The tunable parameters were selected from seven physics schemes of the WRF model. Ten tropical cyclones from different categories over the BoB between 2015 and2018 were considered for the numerical experiments. Three sensitivity analysis methods, namely Morris one at a time (MOAT), the multivariate adaptive regression splines (MARS), and the surrogate-based Sobol' were employed for carrying out the sensitivity experiments. The Gaussian Process Regression (GPR) based Sobol' method produced better quantitative results with 200 samples. The parameter P14 (scattering tuning pa-

rameter used in the shortwave radiation) was seen to influence most of the output variables strongly. The variables surface air pressure (SAP) and downward longwave radiation flux (DLWRF) were found sensitive to most of the parameters. Out of the total selected parameters, eight parameters (P14 - scattering tuning parameter, P6 - multiplier of entrainment mass flux rate, P17 - multiplier for the saturated soil water content, P10 - scaling factor applied to icefall velocity, P4 - Von Karman constant, P22 - profile shape exponent for calculating the momentum diffusivity coefficient, P3 - scaling related to surface roughness,

and P15 - diffusivity angle for cloud optical depth) were found contributing to $80\% - 90\%$ of the total sensitivity metric. A comparison of the WRF simulations with the default and that with optimal parameters with respect to observations showed a 19.65% improvement in the surface wind prediction, 6.5% improvement in the surface temperature prediction, and a 13.3% improvement in the precipitation prediction when the optimal set of parameters is used instead of the default set of parameters. These results indicate that the optimization of model parameters using advanced optimization techniques can further improve

the prediction of tropical cyclones in the Bay of Bengal.

*Code and data availability.* The source code of the WRFv3.9.1 is available at https://github.com/NCAR/WRFV3/releases/tag/V3.9.1 (last access: August 2017). The UQ-PyL software is available at http://www.uq-pyl.com (last access: October 2015). The FNL reanalysis data with a resolution of $1° × 1°$ is available at https://rda.ucar.edu/datasets/ds083.2/index.html#!description and the data with a resolution of $0.25° × 0.25°$ is available at https://rda.ucar.edu/datasets/ds083.3/index.html#!description. The ERA5 reanalysis pressure level data is avail-

able at https://cds.climate.copernicus.eu/cdsapp#!/dataset/reanalysis-era5-pressure-levels?tab=overview and the surfacelevel data is available at https://cds.climate.copernicus.eu/cdsapp#!/dataset/reanalysis-era5-single-levels?tab=overview. The IMDAA reanalysis data is downloaded from https://rds.ncmrwf.gov.in/datasets. The IMERG rainfall data is provided by the NASA GSFC at https://disc.gsfc.nasa.gov/datasets/GPM_3IMERGDF_06/summary. Additionally, the namelist files used for the WRF model simulations, the WRF simulation performed with the default and optimal parameter values, the ncl scripts used to plot the results, and the IPython notebook codes used for the

sensitivity analysis and machine learning algorithms are available at https://doi.org/10.5281/zenodo.5105285(Baki et al., 2021). Though the complete 5000 simulations data is not provided due to the large size, it will be provided on demand.



*Author contributions.* BH and SC conceptualized the goals and designed the methodology. BH performed the data curating and corresponding formal analysis, conducted the investigation. All authors contributed to the interpretation and discussion on results. BH prepared the original draft and SC performed draft editing. The final manuscript is prepared under the supervision of CB and BS.

*Competing interests.* The authors declare no conflict of interest

*Acknowledgements.* The authors are thankful to the international WRF community for their tremendous effort to develop the WRF model. The model simulations are performed on Aqua High-Performance Computing (HPC) system at the Indian Institute of Technology Madras (IITM), Chennai, India, and the Aaditya HPC system at the Indian Institute of Tropical Meteorology (IITM), Pune, India. The FNL data are provided by NCEP, and the IMERG precipitation data are provided by Goddard Earth Sciences Data and Information Services Center.

Authors gratefully acknowledge NCMRWF, Ministry of Earth Sciences, Government of India, for IMDAA reanalysis data. The authors are also thankful to the international scikit-learn community for developing and providing the APIs. The figures are plotted using the python language and NCAR Command Language (NCL).





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





**List of Tables**





**Table 1.** Overview of the adjustable parameters and corresponding ranges selected in this study.

| Index | Scheme | Parameter | Default | Range | Description |
|-------|--------|-----------|---------|-------|-------------|
| P1 | Surface layer | xka | 2.40E-05 | [1.2e-5 5e-5] | The parameter for heat/moisture exchange coefficient (s m-2) |
| P2 | | czo_fac | 0.0185 | [0.01 0.037] | The coefficient for converting wind speed to roughness length over water |
| P3 | | znt_zf | 1 | [0.5 2] | Scaling related to surface roughness |
| P4 | | karman | 0.4 | [0.35 0.42] | Von Kármán constant |
| P5 | Cumulus | pd | 1 | [0.5 2] | The multiplier for downdraft mass flux rate |
| P6 | | pe | 1 | [0.5 2] | The multiplier for entrainment mass flux rate |
| P7 | | ph_usl | 150 | [50 350] | Starting height of downdraft above USL (hPa) |
| P8 | | timec | 2700 | [1800 3600] | Average consumption time of CAPE (s) |
| P9 | | tkemax | 5 | [3 12] | The maximum turbulent kinetic energy (TKE) value in sub-cloud layer (m2 s-2) |
| P10 | Microphysics | ice_stokes_fac | 14900 | [8000 30000] | Scaling factor applied to ice fall velocity (s-1) |
| P11 | | n0r | 8.00E+06 | [5e6 1.2e7] | Intercept parameter of rain (m-4) |
| P12 | | dimax | 5.00E-04 | [3e-4 8e-4] | The limited maximum value for the cloud-ice diameter (m) |
| P13 | | peaut | 0.55 | [0.35 0.85] | Collection efficiency from cloud to rain auto conversion |
| P14 | Short Wave Radiation | cssca_fac | 1.00E-05 | [5e-6 2e-5] | Scattering tuning parameter (m2 kg-1) |
| P15 | Longwave | Secang | 1.66 | [1.55 1.75] | Diffusivity angle for cloud optical depth computation |
| P16 | Land Surface | hksati | 1 | [0.5 2] | The multiplier for hydraulic conductivity at saturation |
| P17 | | porsl | 1 | [0.5 2] | The multiplier for the saturated soil water content |
| P18 | | phi0 | 1 | [0.5 2] | The multiplier for minimum soil suction |
| P19 | | bsw | 1 | [0.5 2] | The multiplier for Clapp and hornbereger "b" parameter |
| P20 | Planetary Boundary Layer | Brcr_sbrob | 0.3 | [0.15 0.6] | Critical Richardson number for boundary layer of water |
| P21 | | Brcr_sb | 0.25 | [0.125 0.5] | Critical Richardson number for boundary layer of land |
| P22 | | pfac | 2 | [1 3] | Profile shape exponent for calculating the momentum diffusivity coefficient |
| P23 | | bfac | 6.8 | [3.4 13.6] | Coefficient for Prandtl number at the top of the surface layer |
| P24 | | cpc_nlfm | 15.9 | [12 20] | Countergradient proportional coefficient of non-local flux of momentum |





**Table 2.** Details of the tropical cyclones selected in this study.

| Cyclone | Landfall time | Simulation duration |
|---|---|---|
| VSCS Thane | 0100 - 0200 UTC 30th Dec 2011 | 2011-12-26_18:00:00 to 2011-12-31_06:00:00 |
| VSCS Phailin | 1700 UTC 12th Oct 2013 | 2013-10-09_06:00:00 to 2013-10-13_18:00:00 |
| VSCS Leher | 0830 UTC 28th Nov 2013 | 2013-11-25_00:00:00 to 2013-11-29_12:00:00 |
| VSCS Madi | 1700 UTC 12th Dec 2013 | 2013-12-09_06:00:00 to 2013-12-13_18:00:00 |
| SCS Helen | 0800 - 0900 UTC 22nd Nov 2013 | 2013-11-19_00:00:00 to 2013-11-23_12:00:00 |
| SCS Mora | 0400 - 0500 UTC 30th may 2017 | 2017-05-26_18:00:00 to 2017-05-31_06:00:00 |
| CS Nilam | 1030 - 1100 UTC 31st Oct 2012 | 2012-10-28_00:00:00 to 2012-11-02_12:00:00 |
| CS Viyaru | 0230 UTC 16th May 2013 | 2013-05-12_18:00:00 to 2013-05-17_06:00:00 |
| CS Komen | 1400 - 1500 UTC 30th July 2015 | 2015-07-27_06:00:00 to 2015-07-31_18:00:00 |
| CS Roanu | 1000 UTC 21st May 2016 | 2016-05-18_00:00:00 to 2016-05-22_12:00:00 |





**Table 3.** RMSE values of variables simulated using default and optimal parameter sets.

| Variable | Default | Optimal | performance improvement % |
|---|---|---|---|
| **WS10 (m/s)** | 2.533963478 | 2.109735527 | 16.74 |
| **SAT (K)** | 2.074763562 | 2.00973412 | 3.13 |
| **SAP (hPa)** | 9.236555992 | 9.169370835 | 0.73 |
| **Precipitation (mm/day)** | 8.363025366 | 7.595082563 | 9.18 |





## List of Figures





**Figure 1.** An illustration of the WRF model configuration with two nested domains.





**Figure 2.** IMD observed tracks of the selected tropical cyclones.





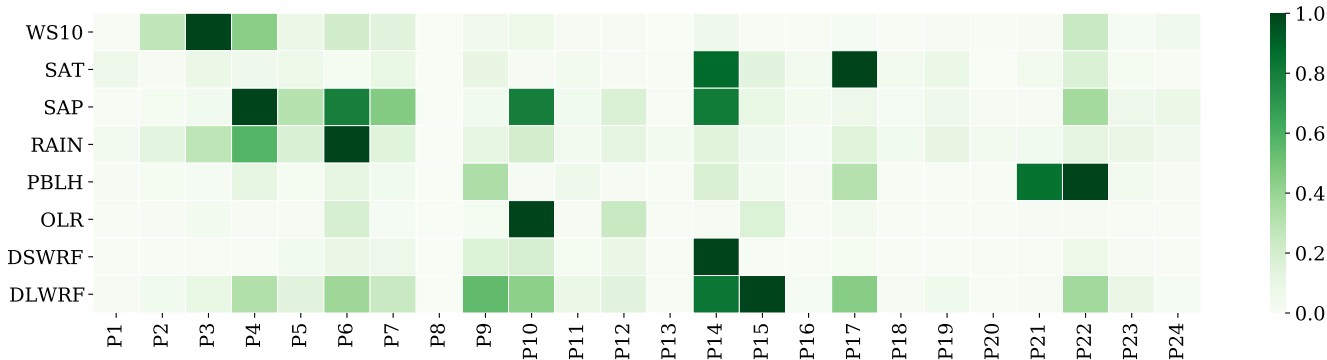

**Figure 3.** The heatmap of the normalized MOAT-modified means of 24 parameters for the meteorological variables considered, with one implying the most sensitive parameter, and zero implying the least sensitive.





**Figure 4.** Box plot of the elementary effects of 24 parameters for the meteorological variables considered. Each dataset is created with 100 applications of bootstrap resampling out of 10 instances. The center lines (red) are the median values; the top and bottom of the boxes are the average ±1 standard deviation; the upper and lower whiskers are the maximum and minimum values.





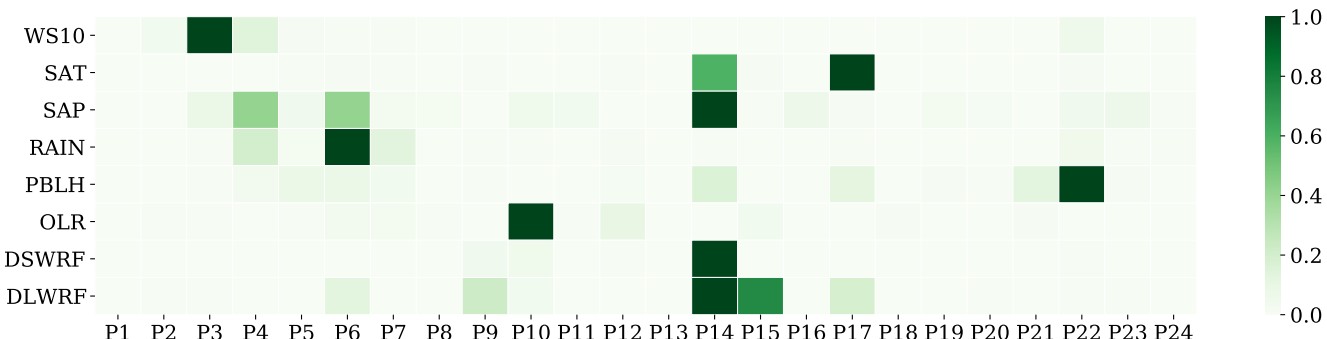

**Figure 5.** Heatmap of the normalized GCV scores of 24 parameters, for the meteorological variables considered, using the MARS method.



**Figure 6.** Box plot of the normalized GCV scores of 24 parameters, for the meteorological variable considered, with the MARS method. Each dataset is created by 100 applications of bootstrap-resampling out of 250 instances.



**Figure 7.** Cross-validation results of the surrogate models GPR, SVM, RF, RT, and KNN, for the meteorological variables considered in the present study.



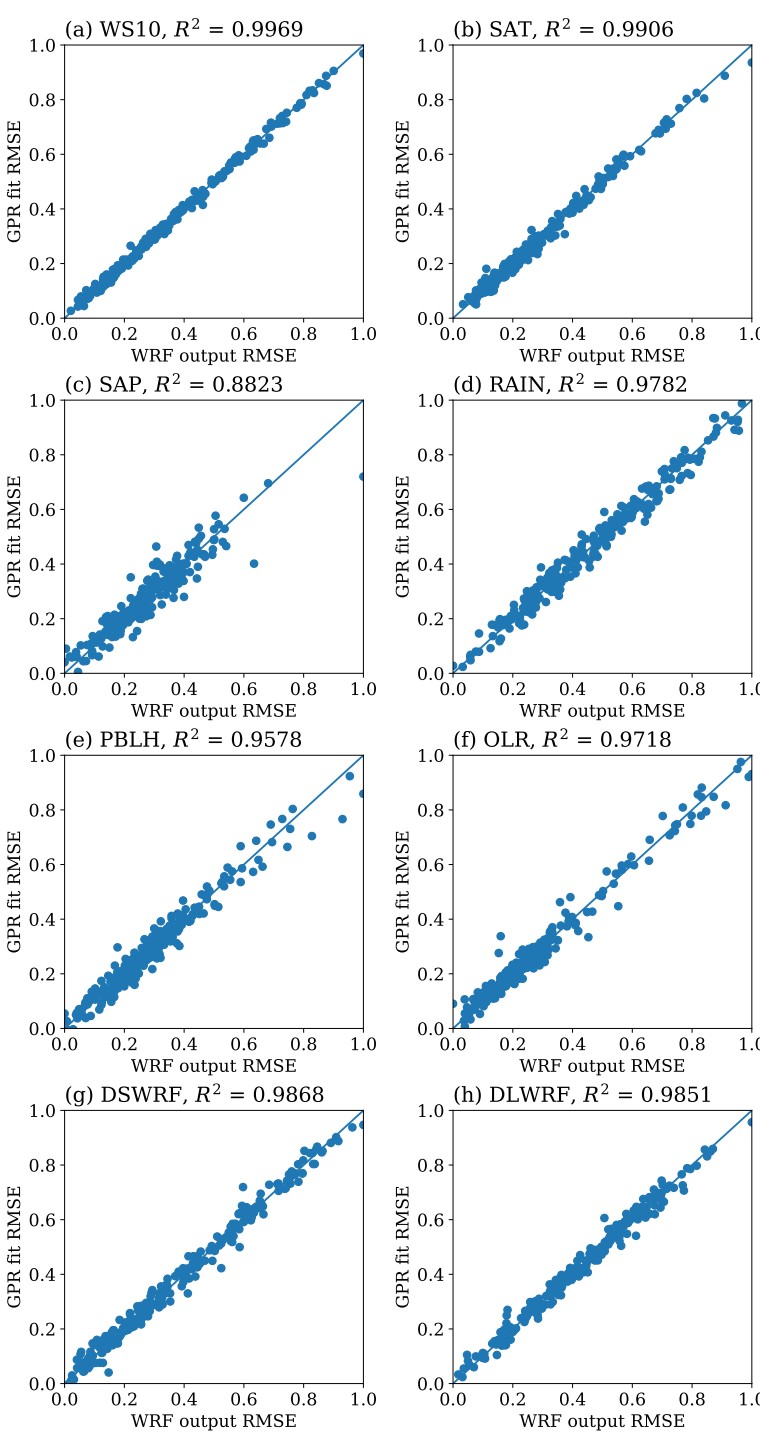

**Figure 8.** Accuracy of the GPR model for a sample size of 250, for the meteorological variables considered. Horizontal axis denotes the RMSE from WRF model and the vertical axis denotes the RMSE from GPR fit.



**Figure 9.** Cross-validation results of the GPR model with different sample sizes of 50, 100, 150, 200, and 250, for the meteorological variables considered.





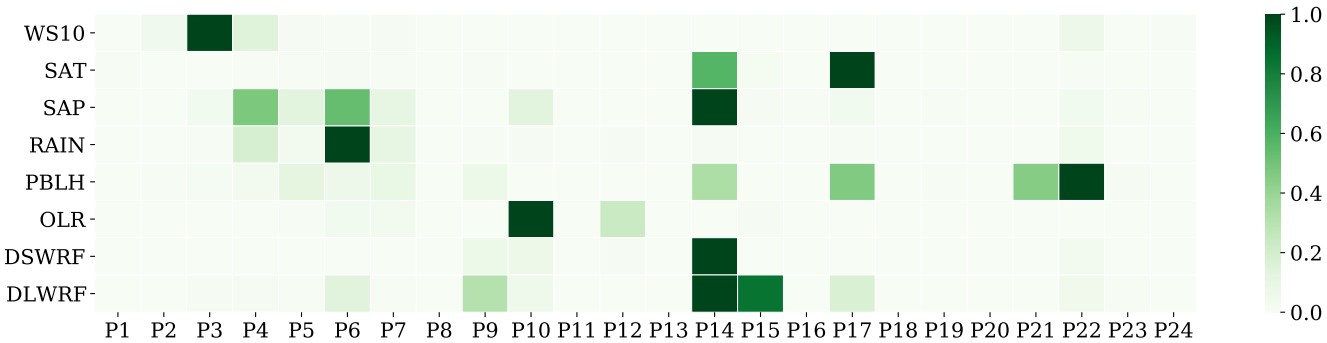

**Figure 10.** Heatmap of the total sensitivity index of 24 parameters for the meteorological variables considered, with the Sobol' sensitivity analysis.



**Figure 11.** Sobol's primary and secondary effects of 24 parameters for the meteorological variables considered.







**Figure 12.** Accumulated relative importance of Sobol' total order effects for different parameters, corresponding to each variable.



**Figure 13.** Ranks of the parameters according to their sensitivities based on (a) MOAT method, (b) MARS method, and (c) Sobol' method.



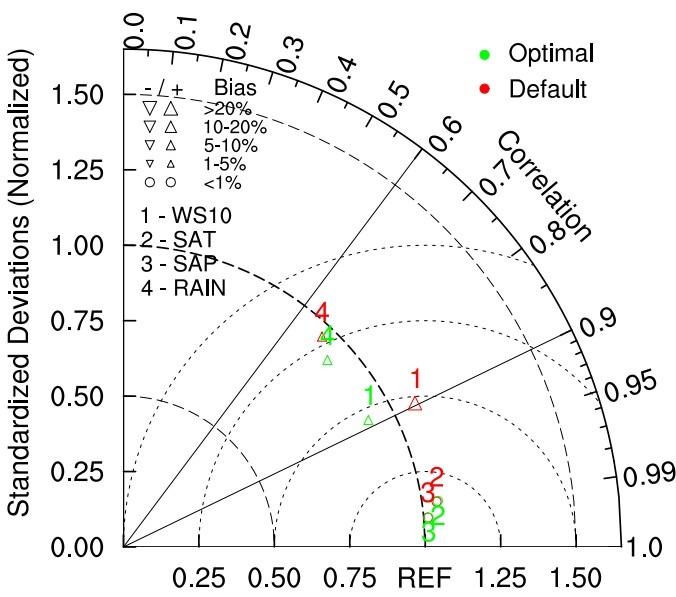

**Figure 14.** Comparison of Taylor statistics of WS10, SAT, SAP, and rain, simulated using the default and optimal parameters, averaged over all the cyclones for the three and half days.

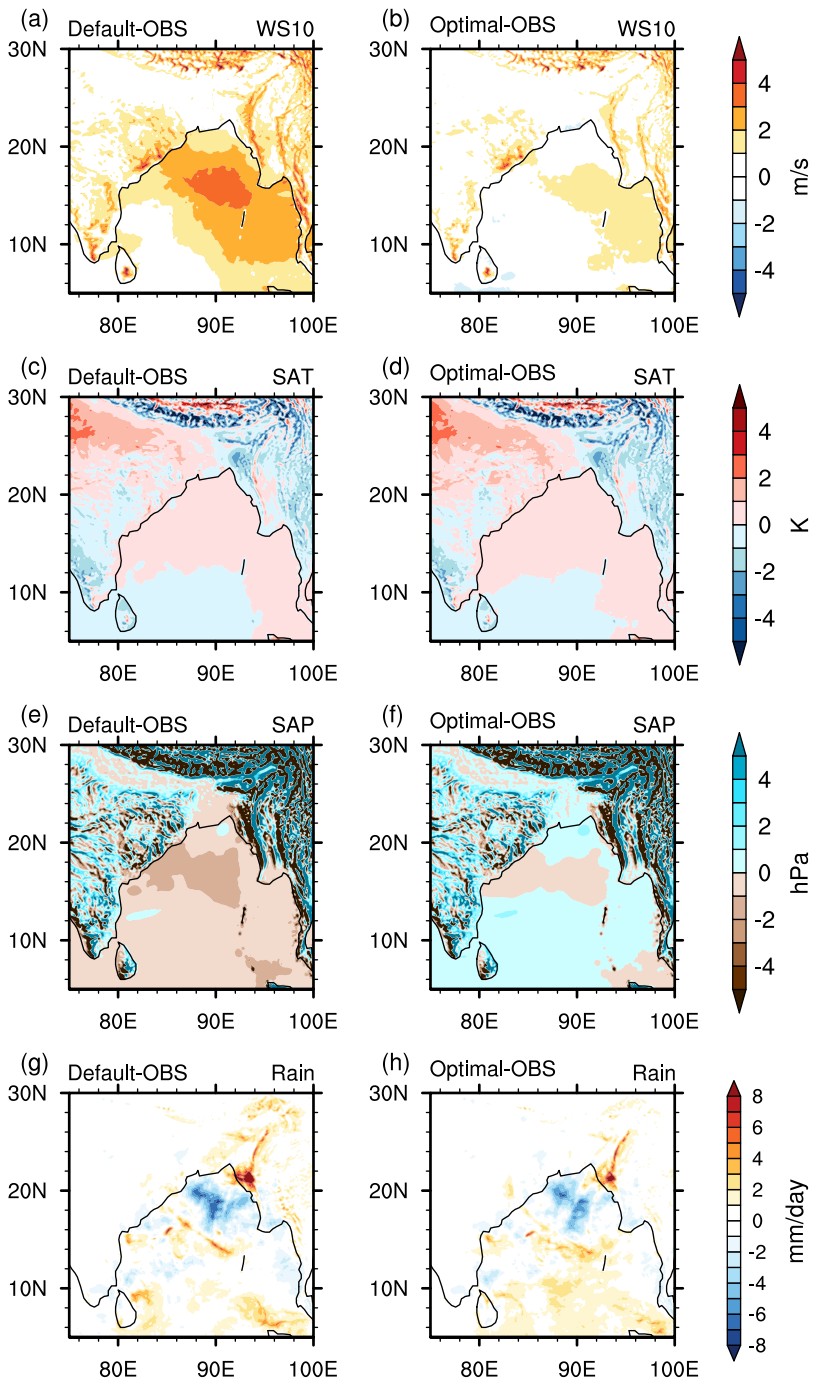

**Figure 15.** Comparison of the spatial distribution of meteorological variables simulated using default and optimal parameters, averaged over all the cyclones for three and half days. Surface wind bias (m/s) (a) between default and observations, (b) between optimal and observations, (c)-(d) surface temperature bias (K), (e)-(f) surface pressure bias (hPa), and (g)-(h) precipitation bias (mm/day).

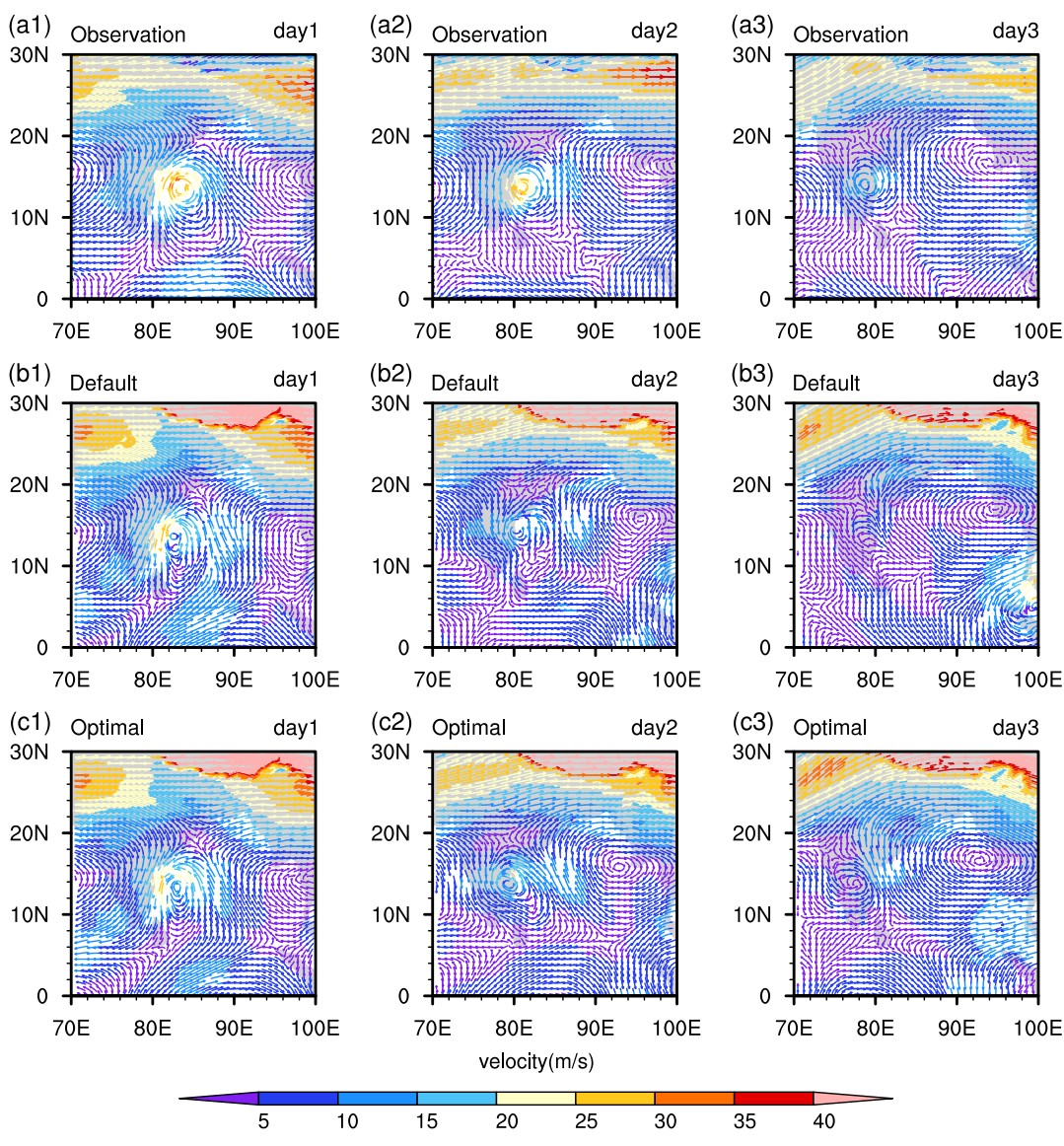

**Figure 16.** The wind velocity field at 500 hPa for the simulation of VSCS Thane using default and optimal parameters, compared with the observations. (a1-a3) show observations at the end of day1, day2, and day3; (b1-b3) show the simulations with default parameters; and (c1-c3) show the simulations with optimal parameters.

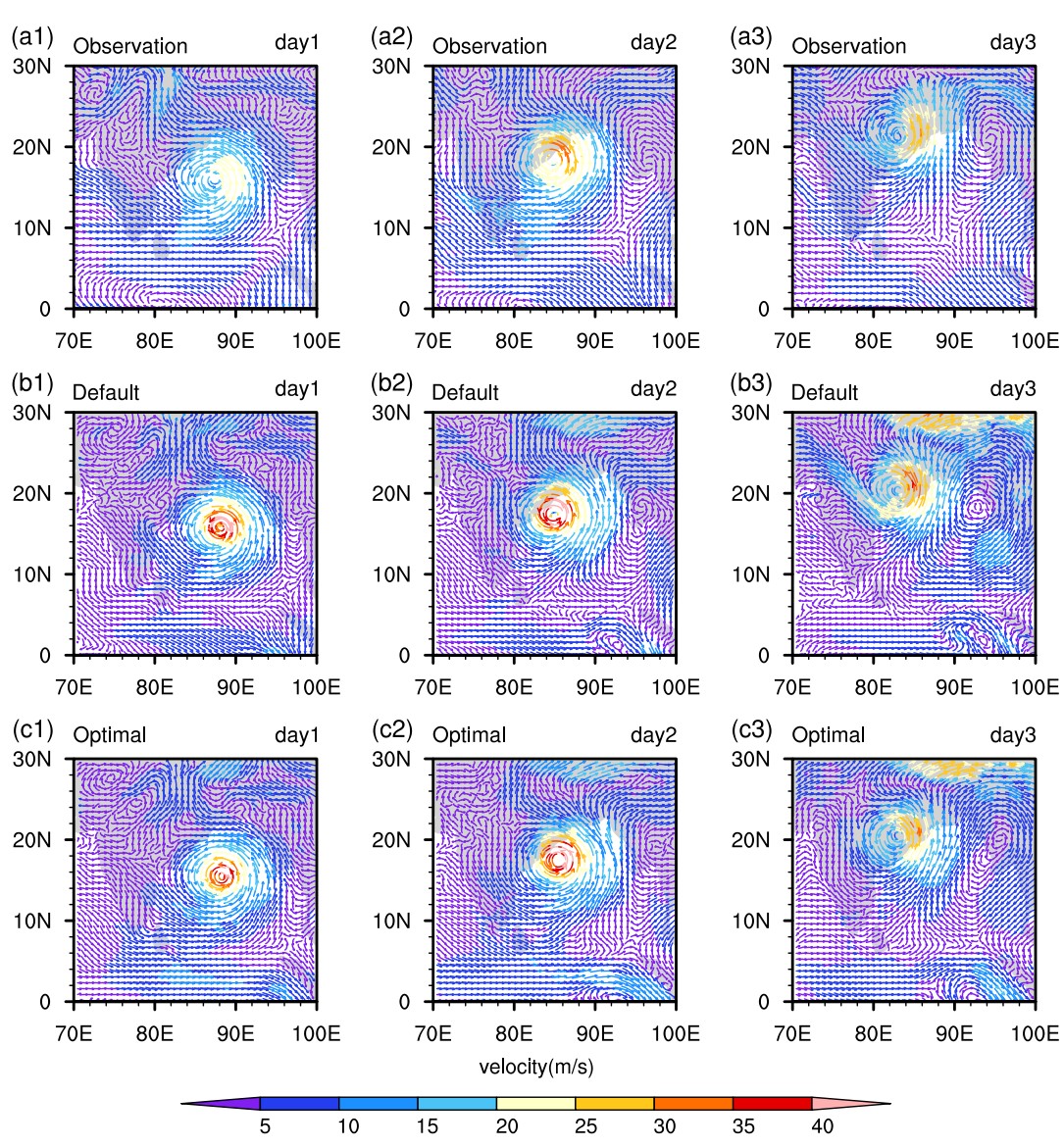

**Figure 17.** Same as Figure 16, for VSCS Phailin

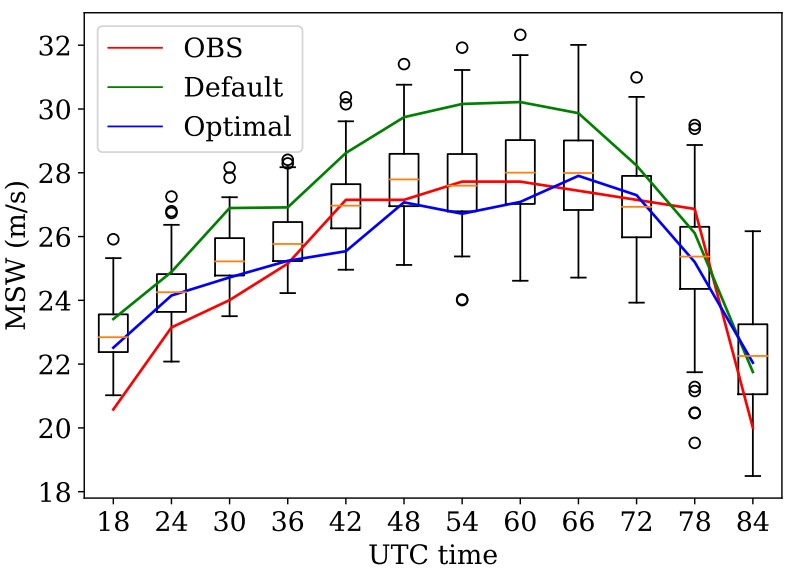

**Figure 18.** Comparisons of three and a half days maximum sustained wind speed averaged over all cyclone simulations using the WRF model with the default and the optimal parameters. The MSW of all cyclones at corresponding forecast hours are averaged, using which the boxplots are generated. The green line shows the simulation with default parameters, the blue line shows the simulations with optimal parameters, and the red line shows the observed MSW. The data is collected at 6 hourly interval and is plotted accordingly.