# Peer review of "Determining the sensitive parameters of WRF model for the simulation of tropical cyclones in the Bay of Bengal using Global Sensitivity Analysis and Machine Learning"

_Geoscientific Model Development, 2021_

## Author Comment (AC1)

**Reply to the Comments by Referee #1 for Manuscript gmd-2021-242 "Determining the sensitive parameters of WRF model for the simulation of tropical cyclones in the Bay of Bengal using Global Sensitivity Analysis and Machine Learning"**

**General comments:**

This study investigated the impacts of 24 tunable parameters in the Weather Research and Forecasting model on the simulations of tropical cyclones over the Bay of Bengal region. Three global sensitivity analysis methods were employed and compared. The parameter sensitivity results were found to be consistent across three methods for all the variables, and 8 out of the 24 parameters contribute 80%−90% to the overall sensitivity scores. Compared to default parameters, applying optimal parameters produced remarkable improvements in the simulated 10m wind speed, surface air temperature, surface air pressure, and precipitation predictions. I think the manuscript is well organized and the presentation is generally good. However, there are some aspects need to be improved before considering of publication.

*The authors appreciate the positive and valuable comments by the referee, which helped in improving the quality of the manuscript. The manuscript has been revised following the referee's comments. A point-by-point response to the comments is provided below.*

**Minor comments:**

**Comment 1:** The word "prediction" is used in the title and in the main text extensively. Please note that the meanings of "prediction" and "simulation" are not exactly the same, and improved simulation with a better model does not always translate into increases in prediction skills. One good example was given by Liu et al. (2019), who showed that the parameters' impacts on simulation and prediction might be different. I understand that the topic of this study is

"simulation", so I suggest replacing the word "prediction" by simulation in the title and in the text.

*Reply 1: The authors thank the reviewer for pointing out. The word "prediction" has been replaced with "simulation" in the revised manuscript text as well as in the title.*

**Comment 2:** Several literatures that are highly related to the selection of parameters are missing in the manuscript. For example, P6 - multiplier of entrainment mass flux rate, P4 - Von Karman constant, and P3 - scaling related to surface roughness, which are found to be important for tropical cyclone simulations in this study, were primarily identified by Yang et al. (2012) and Yang et al. (2017). These papers should be cited accordingly

*Reply 2: Point well taken. The following citations have now been added in the introduction part of the revised manuscript.*

*Yang et al., (2012) conducted an uncertainty quantification and tuning of five key parameters found in the new Kain-Fritsch scheme of the WRF model, using the Multiple Very Fast SImulated Annealing (MVFSA) sampling algorithm. The authors have reported that the optimal parameters reduced the model precipitation bias significantly, and the model performance is sensitive to the downdraft and entrainment related parameters. Yang et al., (2017) studied the sensitivity of 25 parameters within the Mellor-Yamada-Nakanishi-Niino (MYNN) planetary boundary layer scheme and MM5 surface layer scheme of the WRF model, for the simulations of turbine height wind speed, and reported that more than 60% of the output variance is contributed by only 6 parameters.*

*Yang et al. (2012): Some issues in uncertainty quantification and parameter tuning: a case study of convective parameterization schemes in the WRF regional climate model, Atmos. Chem. Phys., 12:2409-2427*

*Yang et al. (2017): Sensitivity of Turbine-Height Wind Speeds to Parameters in Planetary Boundary-Layer and Surface-Layer Schemes in the Weather Research and Forecasting Model, Boundary-Layer Meteorology. 162:117–142*

**Specific comments:**

**Comment 1:** Line 24, ".Singh et al. (2021a)."?

**Reply 1:** *The punctuation mark before the author has been removed in the revised manuscript, and ".Singh et al. (2021a)." is changed to "Singh et al. (2021a)."*

**Comment 2:** Line 25, "Singh et al. (2019) showed that present warming climate impacts on the …", please check the grammar.

**Reply 2:** *The mistake has been rectified in the revised manuscript. The sentence is changed to "Singh et al.,(2019) showed that the present warming climate impacts the formation and severity of the tropical cyclones over the BoB region"*

**Comment 3:** Line 29, What does "VSCS" mean?

**Reply 3:** *VSCS is the short form of Very Severe Cyclonic Storms. The expansion has been provided in the revised manuscript.*

**Comment 4:** Line 50, "at once" -> "simultaneously"?

**Reply 4:** *At line 50,"at once" is changed to "simultaneously" in the revised manuscript.*

**Comment 5:** Line 108, "in question to"?

**Reply 5:** *At line 108, "caused by the variable in question" is changed to "caused by that variable" in the revised manuscript.*

**Comment 6:** Line 395-402, the definition of P6 and the analyses about its impacts largely follows that of Yang et al. (2012), which should be added here. Meanwhile, it is not clear to me why suppressed convection (i.e. weakened consumption of CAPE or instability) leads to more "stable" stratiform clouds. Have the authors checked the vertical profiles of atmosphere temperature and moisture? One explainable for the changes in stratiform precipitation is the competition for moisture between convective and stratiform processes as indicated by Liu et al. 2018.

*Reply 6: The authors thank the reviewer for his valuable suggestions. Though the vertical profiles of atmospheric temperature and moisture were not examined in the current study, the explanation to the above mentioned statement is found through the studies of Yang et al.,(2012) and Liu et al.,(2018). The citations are added in the revised manuscript as follows.*

*The parameter P6 is the entrainment of mass flux rate in the Kain-Fritsch cumulus physics scheme, which has been identified as a sensitive parameter for the simulations of precipitation in the studies of Yang et al.,(2012). The entrainment of air into the updrafts indicates a detrainment of moisture from the updrafts, which is the key water source for the formation of stratiform clouds. This indicates that the formation of stratiform clouds compensates for the reduction of convective processes and leads to an increase in the stratiform precipitation (Liu et al., 2018).*

*Liu et al. (2018): Combined impacts of convection and microphysics parameterizations on the simulations of precipitation and cloud properties over Asia, Atmospheric Research, 212:172-185*

---

## Author Comment (AC2)

**Reply to the Comments by Referee #2 for Manuscript gmd-2021-242 "Determining the sensitive parameters of WRF model for the prediction of tropical cyclones in the Bay of Bengal using Global Sensitivity Analysis and Machine Learning"**

**Referee #2**

The study investigates the sensitivity of 8 meteorological variables to 24 WRF model parameters. Three methods are being used for the sensitivity analysis, with very similar results, indicating that the results are robust. Most variables are only sensitive to a few parameters, and some parameters don't introduce any obvious sensitivity. The results can help to improve forecasting by finding optimal tuning parameters in numerical models.

The study is well-constructed and generally well written. I don't see any major flaws that would prohibit publication. In fact, I think this is a nice systematic study that can tell us a lot about how to improve forecast models, and how to find the optimum value for the myriad of tunable model parameters. I do have some concerns, but they don't really apply to the methodology or interpretation of results. Please see below.

*The authors appreciate the positive and valuable comments by the referee, which helped in improving the quality of the manuscript. The manuscript has been revised following the referee's comments. A point-by-point response to the comments is provided below.*

**Major comments:**

**Comment 1:** I feel there is a contradiction between the title and motivation of this study versus the presented results. The title and motivation of the study explicitly refer to TCs, but most of the results are not TC specific. Rather, the results presented in Figs. 3–15 seem to be derived from the entire domain 2, of which TCs only cover a small fraction of. So, to me it seems the results are general rather than specific to TCs (note that I don't think this is a bad thing). Only Figs. 16–18 specifically refer to TCs. This contradiction could be removed by either focusing the analysis to parts of the domain that include the TC (like the panels in Figs. 16 and 17) or by rewording and restructuring the title and text.

**Reply 1:** *Point well taken. The authors agree with the referee that the results were obtained based on simulations conducted over the domain that surrounds the Bay of Bengal. The parameter sensitivity generally depends on local conditions, and the type of events simulated (Di et al., 2015; Quan et al., 2016). In the present study, tropical cyclones over the Bay of Bengal were chosen for the sensitivity experiments, and ten cyclones across different categories were simulated using the WRF model for generalizing the outcomes. The selected cyclones originated at the center of the Bay of Bengal region and had landfall at different locations. In addition, these cyclones are simulated for a time period of 108 hours, which requires a huge computational domain to cover the entire life cycle and to include the influences of adjoining area along the cyclone track. By considering all these factors, one big domain is considered for the numerical simulations instead of different domains for each cyclone. Though the model simulations were averaged over the entire domain, the results do strongly depend on the type of simulation event too, in this case, tropical cyclones. So, the authors chose to have both the type of the event simulated (tropical cyclones) and the region of interest (Bay of Bengal) in the title.*

Di, Z., Duan, Q., Gong, W., Wang, C., Gan, Y., Quan, J., Li, J., Miao, C., Ye, A., and Tong, C.: Assessing WRF model parameter sensitivity:A case study with 5 day summer precipitation forecasting in the Greater Beijing Area, Geophysical Research Letters, 42, 579–587, 2015.

Quan, J., Di, Z., Duan, Q., Gong, W., Wang, C., Gan, Y., Ye, A., and Miao, C.: An evaluation of parametric sensitivities of different meteorological variables simulated by the WRF model, Quarterly Journal of the Royal Meteorological Society, 142, 2925–2934, 2016.

**Comment 2:** Somewhat related to the first comment, the authors seem to treat all variables with the same importance. Again, I don't think this is a bad thing, but if this study is about TCs, I'd put TC-specific variables, such as 10-m winds and rainfall (and maybe pressure), in the focus. Again this could be done by restructuring the text.

**Reply 2:** *Point well taken. The surface wind speed, surface pressure, and rainfall are indeed the predominant variables for tropical cyclone simulations, and several studies (Di et al., 2020) were also conducted particularly for cyclonic events (typhoons in the mentioned study) by focusing on these variables only. However, Quan et al., (2016) reported that the atmospheric variables such as outgoing longwave radiation (OLR), planetary boundary layer height (PBLH), and downward shortwave and longwave radiation fluxes( DSWRF, DLWRF) were also exerting considerable influence on precipitation. In addition, surface temperature plays a significant role in forming cyclones and their intensification. Taking these into account, the present study considered eight model output variables, rather than just wind speed and precipitation, with the same importance. Our studies also found that six out of eight sensitive parameters were exerting a significant influence on surface wind speed, surface pressure, surface temperature, and precipitation, which are the predominant variables for cyclones, as discussed. This indicates that no parameter that is influencing surface wind speed, surface pressure, surface temperature, and precipitation, has been left out due to the consideration of extra variables.*

Di, Z., Duan, Q., Shen, C., and Xie, Z.: Improving WRF typhoon precipitation and intensity simulation using a surrogate-based automatic parameter optimization method, Atmosphere, 11, 89, 2020.

**Comment 3:** How did you decide on the ranges in Table 1?
**Reply 3:** *In the present study, the selected parameters and their ranges have been adopted from the studies of Di et al., (2020), who conducted a parameter sensitivity and calibration for*

*25 parameters to improve the precipitation and intensity simulations of Typhoons over South China. Similar studies were also conducted by researchers Di et al., (2015) and Quan et al., (2016), who examined the importance of the 23 parameters for the simulations of summer precipitation events over the Greater Beijing area. From these studies, the parameter default values and their ranges were identified for the selected parameters.*

**Comment 4:** L87: Unless there's a technical reason for not doing so, to me it makes more sense to say "The objective of the present study is to assess the sensitivity of meteorological variables such as surface pressure, temperature, wind speed, precipitation, to WRF model parameters..." instead of "The objective of the present study is to assess the sensitivity of the WRF model parameters to various meteorological variables such as surface pressure, temperature, wind speed, precipitation, ...". I think what we're interested in is the response of the output to the input, and the latter sounds to me as the opposite.

**Reply 4:** *The authors thank the reviewer for point this out. The sentence has been updated in the revised manuscript as follows.*

*The objective of the present study is to assess the influence of the WRF model parameters on various meteorological variables such as surface pressure, temperature, wind speed, precipitation, and model variables such as radiation fluxes and boundary layer height, for the simulations of tropical cyclones over the BoB region, using three different global sensitivity analysis methods.*

*Similar changes have been done throughout the revised manuscript.*

**Comment 5:** Fig. 18: I wonder if we're losing some information by showing boxplot aggregates and the average wind speed of all TCs. Often the average is something non-physical and often does not tell us much. Would it be possible to show a ten time series, one for each TC? Also, how do the box plots relate to the colored lines? Shouldn't the green or blue line go right through the orange lines in the box plots?

**Reply 5:** *The time series boxplots of maximum sustained wind speed (MSW) obtained from the simulations conducted for the MARS and Sobol' sensitivity studies are plotted for individual cyclones, as shown in Figure 1. A corresponding description has been added in the revised manuscript.*

[Figure]

Figure 1: Comparisons of three and a half days maximum sustained wind speed (MSW) OF all cyclone simulations using the WRF model with the default and the optimal parameters. The boxplots of individual cyclones are obtained from the 250 simulations used for the MARS and Sobol' analysis. The green line shows the simulation with default parameters, the blue line shows the simulations with optimal parameters, and the red line shows the observed MSW. The data is collected at 6 hourly interval and is plotted accordingly.

*In the boxplots, the blue line indicates the observations, whereas the red and green lines indicate the default and optimal simulations, respectively. It is to be noted that the orange line in the boxplot indicates the mean of the MSW from the 250 simulations, and the boxplot represents the variability of the model simulations with respect to varying model parameters. In contrast, the default or optimal simulations need not to exactly pass through the mean, but should lie within the limits of the boxplot. The same can be observed in the figure.*

**Specific comments:**

**Comment 1:** L. 21: "which alone contributed to an overall increase in the NIO." – Increase of what? Activity or destructiveness?

**Reply 1:** *The sentence has been changed to "The frequency and duration of very severe cyclones in the BoB were increasing at an alarming rate, which alone contributed to an overall increase in the frequency over the NIO", in the revised manuscript.*

**Comment 2:** L 59: It would be interesting to know which two parameters were found to significantly affect the intensity and structure.

**Reply 2:** *Green and Zhang (2014) conducted a sensitivity study to examine the influence of four parameters related to the fluxes of momentum and moist enthalpy across the air-sea interface. The four parameters $\alpha, V_c, m,$ and $\beta$ are selected from the drag coefficient ($C_D$) and moist enthalpy coefficients equations. The authors reported that the multiplication factors $\alpha$ and $\beta$ control the intensity and structure of the tropical cyclnes at a greater extent. The same has been updated in the revised manuscript.*

Green, B. W., & Zhang, F. (2014). Sensitivity of tropical cyclone simulations to parametric uncertainties in air–sea fluxes and implications for parameter estimation. Monthly Weather Review, 142(6), 2290-2308.

**Comment 3:** L194: "spin-up time" instead of "spin-off time" (also L215)

**Reply 3:** *The sentences have been updated with "spin-up" in the revised manuscript.*

**Comment 4:** L444: Taylor statistics are...

**Reply 4:** *The sentence has been updated with "Taylor statistics are" in the revised manuscript.*

**Comment 5:** L446: can be plotted in one Taylor diagram...

**Reply 5:** *The sentence has been updated with "can be plotted in one Taylor diagram" in the revised manuscript.*

**Comment 6:** To me, there is a discrepancy between Figs. 14 and 15. In Fig 14, it doesn't look like the optimal parameters are any better than the default parameters when looking at WS10. But when looking at Figs. 15a and 15b, it looks like the optimal parameters are quite a bit better (smaller bias). How do you explain this discrepancy?

**Reply 6:** *In the Taylor diagram (Figure 14), the differences between the simulations with the default and optimal parameter values are explained with four statistics, which are: centered RMS error, correlation, normalized standard deviation, and bias. The first three statistics can be explainable in the plot itself, where as the bias cannot be explained inside the plot. Thus, it is represented with different sizes of markers. In the Figure 14, it can be seen that the bias value corresponding to the optimal parameters is $5 - 10\%$, whereas the that of the default parameters is $10 - 20\%$. The same is confirmed by the higher bias for default parameters in domain averaged surface plot (Figure 15).*